# Cell Proliferation PET Imaging with 4DST PET/CT in Colorectal Adenocarcinoma and Adenoma

**DOI:** 10.3390/diagnostics11091658

**Published:** 2021-09-10

**Authors:** Ryogo Minamimoto, Hisako Endo

**Affiliations:** 1Division of Nuclear Medicine, Department of Radiology, National Center for Global Health and Medicine, Tokyo 1628655, Japan; 2Department of Clinical Pathology, Edogawa Hospital, Tokyo 1330052, Japan; hendo4638@yahoo.co.jp

**Keywords:** 4DST PET/CT, colorectal adenoma, colorectal adenocarcinoma, FDG PET/CT, cell proliferation imaging

## Abstract

An age of 70-year-old man was incidentally found two focal high 2-[18F]-fluoro-2-deoxy-d-glucose (FDG) uptake in the descending colon and in the sigmoid colon. We observed the feature of these two areas in the preplanned 4′-[methyl-11C]-thiothymidine (4DST) positron emission tomography (PET)/computed Tomography (CT)providing cell proliferation imaging. A mass forming high 4DST uptake in the descending colon and focal moderate 4DST uptake in the sigmoid colon was confirmed, and that were proven pathologically as adenocarcinoma and moderate to severe type tubular adenoma, respectively. This is the first report to present that colorectal adenoma can be visualized by proliferation PET imaging and the degree of uptake may enable discrimination of colorectal adenoma from adenocarcinoma, based on pathological considerations.

Colon adenomas are regarded as precursors of colorectal cancer (CRC) [1], and the adenoma–carcinoma sequence is a widely accepted theory [2]. Detecting and removing advanced colon adenomas are significant aspects of prevention of CRC [3]. CRC can develop over several years from precancerous adenomatous polyps; however, as most colon adenomas will not progress to carcinoma, the growth and the transition rates of adenomas are unpredictable. Therefore, definitive noninvasive imaging enables differentiation of lesion status between CRC and colon adenoma.

The usefulness of FDG PET and PET/CT for colon adenoma detection has been previously reported [4,5,6]; the sensitivity of FDG-PET depends on size and histologic grade of the colon adenoma [4]. Although FDG-PET has a high potential as an examination to detect colon adenomas that should be removed, the cut-off value of FDG uptake for differentiating malignant from benign lesions tends to be a high maximum standardized uptake value (SUVmax) [7]. Moreover, the FDG uptake pattern of cancer has much in common with that of colon adenoma, resulting in difficulty differentiating between CRC and colon adenoma [8].

A PET tracer for cell proliferation imaging, known as 4DST, is incorporated into DNA [9]. A previous study has reported a higher correlation of proliferating lung cancers and renal cell cancers with 4DST than FDG uptake [10,11]. The potential of 4DST PET/CT for identification of advanced CRC has been reported, with uptake values lower than those for FDG, which shows a similar tendency to the present case [12]. We found that tubular adenoma could be visualized by 4DST-PET/CT (SUV_max_ 2.5) as well as by FDG (SUV_max_ 7.7) (Figure 1). In addition, we demonstrated the potential of proliferation PET for discriminating between CRC and colon adenoma. To clarify the significance of 4DST uptake in the colon adenoma, we compared the 4DST and FDG findings with respect to the pathological results (Figure 2).

Güreşci et al. reported a significant correlation of Ki-67 index with SUV_max_ of FDG uptake in colorectal lesions including CRC and colon adenoma [13]. Moreover, higher proliferation is related to poorer outcomes in CRC [14]. In contrast, among the various types of cancer, the correlation between GLUT-1 expression and SUV_max_ is lowest in colorectal cancer (r = 0.21) [15], which indicates that several factors, other than GLUT-1 expression, may influence FDG uptake in CRC. The expression of Ki-67 shows clear differences between advanced and nonadvanced adenomas [16]. The strong reactions of Ki-67 in adenomas with severe dysplasia show a close association with colorectal carcinoma [17]. GLUT-1 immunostaining was absent in normal colorectal epithelium and tubular adenomas, and absent or only weakly apparent in tubulovillous adenomas [18]. In contrast, GLUT-1 expression in colon adenoma has been reported in several studies. Greijer et al. concluded that HIF-1α expression led to the expression of downstream target genes, such as GLUT-1 [19]. de Wit et al. found that that plasma membrane staining of GLUT-1 was completely absent in low-risk colon adenomas and present in high-risk colon adenomas (21.4%), which is significantly lower than that in CRCs (42.5%) [20]. In the present case, Ki-67 appeared to relate with 4DST and FDG uptake, whereas GLUT-1 expression was low in colon adenoma specimens even in the case of high FDG uptake. This finding suggests that proliferation imaging may have potential for predicting the growth of colon adenomas and their transition to CRC.

## Figures and Tables

**Figure 1 diagnostics-11-01658-f001:**
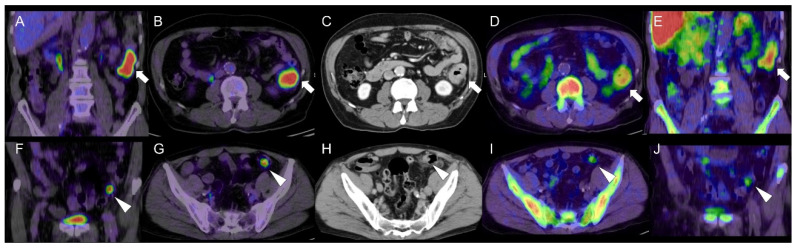
A 70-year-old man with pathologically proven adenocarcinoma in the descending colon and moderate to severe type tubular adenoma in the sigmoid colon. An adenocarcinoma is seen in the descending colon on (**A**) coronal- and (**B**) axial-fused FDG PET/CT imaging (arrows); (**C**) contrast-enhanced CT (arrow); and (**D**) axial- and (**E**) coronal-fused 4DST PET/CT imaging (arrows). The colon adenoma is seen on (**F**) coronal- and (**G**) axial-fused FDG PET/CT (arrowheads); (**H**) contrast-enhanced CT (arrowhead); and (**J**) axial- and (**I**) coronal-fused 4DST PET/CT imaging (arrowheads). FDG PET/CT revealed high FDG uptake (SUV_max_ 15.2) in the advanced CRC at the descending colon, and the lesion was also confirmed with high 4DST uptake (SUV_max_ 4.8). We found that tubular adenoma could be visualized by 4DST-PET/CT (SUV_max_ 2.5) as well as by FDG (SUV_max_ 7.7).

**Figure 2 diagnostics-11-01658-f002:**
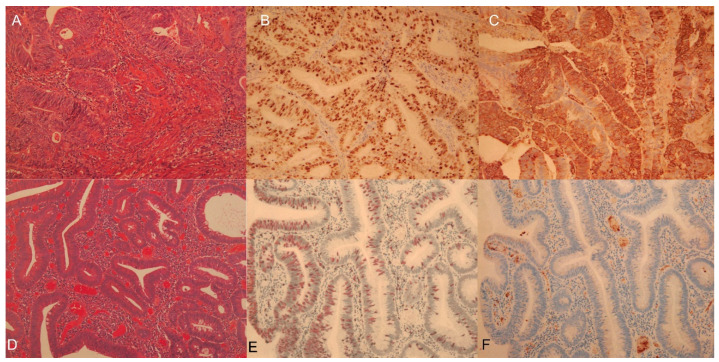
Pathological features of colorectal cancer and adenoma. (**A**) Hematoxylin and Eosin (HE) staining in CRC (moderately differentiated tubular adenocarcinoma), (**B**) Ki-67 staining in CRC (Ki-67 index: 84%), (**C**) GLUT-1 staining in CRC, (**D**) HE staining in colon adenoma (tubular adenoma with moderate to severe atypia), (**E**) Ki-67 staining in colon adenoma (Ki-67 index: 23%), and (**F**) GLUT-1 staining in colon adenoma. Immunohistochemical expression of GLUT-1 and Ki-67 were clearly confirmed in CRC. In colon adenoma, Ki-67 was less intense but GLUT-1 expression was quite low despite the high FDG uptake.

## Data Availability

Not applicable.

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
