# Peer review of "Cell Proliferation PET Imaging with 4DST PET/CT in Colorectal Adenocarcinoma and Adenoma"

_diagnostics, 2021, doi:10.3390/diagnostics11091658_

Round 1

Reviewer 1 Report

Manuscript entitled "Cell proliferation PET imaging with 4DST PET/CT in colorectal adenocarcinoma and adenoma"

This work is potentially interesting while some modifications should be made before final acceptance:

  1. The quality of pathological figures should be substantially improved. Some are dark and blur.
  2. Is there any follow up image before surgical treatment?

Author Response

We greatly appreciate your review of our manuscript and the helpful suggestions.

  1. The quality of pathological figures should be substantially improved. Some are dark and blur.

We appreciate the reviewer’s instructive comments. We revised the quality of pathological figures.

      2. Is there any follow up image before surgical treatment?

We appreciate the reviewer’s comments. There were no extra images before surgical treatment.

Reviewer 2 Report

This is a well written case report, describing the 4DST and FDG PET scanning cahracteristics, alongside the histology with special stains, of a patient with a simultaneous adenoma and cancer. While perhaps of moderate impact and interest, it is well written and clear. A larger case series would be of more use in charcterising the findings of these lesions on 4DST PET, and special staining.

Author Response

We greatly appreciate your review of our manuscript.

Round 2

Reviewer 1 Report

The authors do nothing for my suggestion